# Causal effect of porphyria biomarkers on alcohol-related hepatocellular carcinoma through Mendelian Randomization

Xiaoyu Yang[1,2], Shuomin Wang[1,2], Chen Sun[1,2], Yunhong Xia[1,2]*

1 Department of Oncology, the First Affiliated Hospital of Anhui Medical University, Hefei, Anhui, China,
2 Department of Oncology, Anhui Public Health Clinical Center, Hefei, Anhui, China

* yhxia21@sina.com

**Data Availability Statement:** All relevant data are within the paper and its Supporting Information files.

## Abstract

### Purpose

According to some cohort studies, an association exists between acute intermittent porphyria (AIP) and liver cancer. However, establishing a definitive causal relationship between porphyria and hepatocellular carcinoma (HCC) remains challenging. Prexisting studies regarding porphyria biomarkers and alcohol-related hepatocellular carcinoma (AR-HCC) make possible an entry point. In this study, we aimed to investigate the causal relationships between biomarkers of two types of porphyria, AIP and congenital erythropoietic porphyria (CEP), and AR-HCC.

### Methods

Single-nucleotide polymorphisms (SNPs) associated with porphobilinogen deaminase (PBGD) and uroporphyrinogen-III synthase (UROS), along with outcome data on AR-HCC, were extracted from public genome-wide association studies (GWAS). The GWAS data were then used to explore the potential causal relationships via a two-sample Mendelian randomization (MR) analysis. The effect estimates were calculated using the random-effect inverse-variance-weighted (IVW) method. Additionally, the Cochrane's Q test, MR-Egger test, and leave-one-out analysis were conducted to detect heterogeneity and pleiotropy in the MR results.

### Results

Using the IVW method as the primary causal effects model in the MR analyses, we found that both PBGD (effect estimate = 1.51; 95% CI, from 1.08 to 2.11, p = 0.016) and UROS (effect estimate = 1.53; 95% CI, from 1.08 to 2.18, p = 0.018) have a significant causal effect on AR-HCC.

### Conclusion

Our findings revealed a causal effect of both PBGD and UROS on AR-HCC, suggesting that both AIP and CEP have a causal association with AR-HCC.

**Funding:** This study was funded by the National Natural Science Foundation of China (No. 81472331), Natural Science Foundation of Anhui Province (No. 2108085MH289), and the Project of Scientific Research Foundation of Anhui Medical University (No. 2019xkj146).

**Competing interests:** The authors have declared that no competing interests exist.

## Introduction

Alcohol-related liver disease is the most prevalent chronic liver disease worldwide, accounting for 30% of hepatocellular carcinoma (HCC) cases and HCC-specific deaths [1, 2]. Patients with alcohol-related hepatocellular carcinoma (AR-HCC) are diagnosed at the late stage and have a poorer prognosis than patients with non-alcoholic HCC [1]. Several treatment options are available, including liver transplantation, surgical resection, percutaneous ablation, radiation, and transarterial and systemic therapies [3], and studies on AR-HCC risk factors are continuously being performed. Some AR-HCC risk factors have been identified that modulate lipid metabolism, oxidative stress, inflammation, and ethanol metabolism [4]. In a U.S. study, 1.5% of patients with acute hepatic porphyrias had HCC [5]. Porphyria is a set of rare metabolic disorders caused by enzyme deficiency in the porphyrin metabolic pathway. The synthesis of hemoglobin and other porphyrin compounds is affected in porphyria. Acute intermittent porphyria (AIP) and congenital erythropoietic porphyria (CEP) are relatively common subtypes of porphyria [6]. AIP is caused by porphobilinogen deaminase (PBGD) haploinsufficiency [7, 8], which leads to episodes of intense pain. In contrast, CEP is caused by an error in heme synthesis due to uroporphyrinogen III synthase (UROS) deficiency. Patients with CEP often experience skin photosensitivity and chronic hemolytic anemia [9, 10]. Both AIP and CEP significantly affect the quality of life of patients [11, 12]. Therefore, studies that aim to enhance the diagnosis and treatment of this disease is of immense importance. Several studies have found a connection between porphyria and other diseases [13–15]. In fact, a possible association between AIP and liver cancer was identified in a cohort study through investigation on porphyria biomarkers [16]. In another cohort study comprising patients with AIP, researchers confirmed a high risk of primary liver cancer and identified a strong association with AIP biomarkers [17]. Results from a case-control study showed that AIP patients drink alcohol regularly, but the intake was significantly lower in the symptomatic cases [18]. PBGD and UROS are important biomarkers of AIP and CEP, respectively. GWAS studies on porphyria biomarkers are still in the emerging stage, however, high-quality data of PBGD and UROS is already available. The GWAS studies on PBGD and UROS are currently limited to the European population. Additionally, the available HCC GWAS data for European populations predominantly focus on AR-HCC [2, 19]. The relationship between other porphyria subtypes and AR-HCC has not been thoroughly investigated. In the present study, we aimed to explore the causal association between porphyria biomarkers and AR-HCC.

Mendelian randomization (MR) is an epidemiological method that employs genetic variants as instrumental variables to deduce the causality of exposure-outcome associations [20]. Unlike conventional observational studies, MR analysis offers advantages in mitigating reverse causation bias as allelic randomization precedes disease onset [21, 22].

In our study, we conducted MR analysis with single-nucleotide polymorphisms (SNPs) that were strongly associated with PBGD and UROS to investigate the causal relationship between these two biomarkers and AR-HCC.

## Materials and methods

### Study design

Fig 1 presents the study design. The three key assumptions of MR are as follows: (A) a reliable correlation exists between the instrument (SNPs) and exposure (PBGD and UROS); (B) SNPs are not correlated with any confounding factors; and (C) the influence of SNPs on AR-HCC is solely mediated through AIP and CEP.

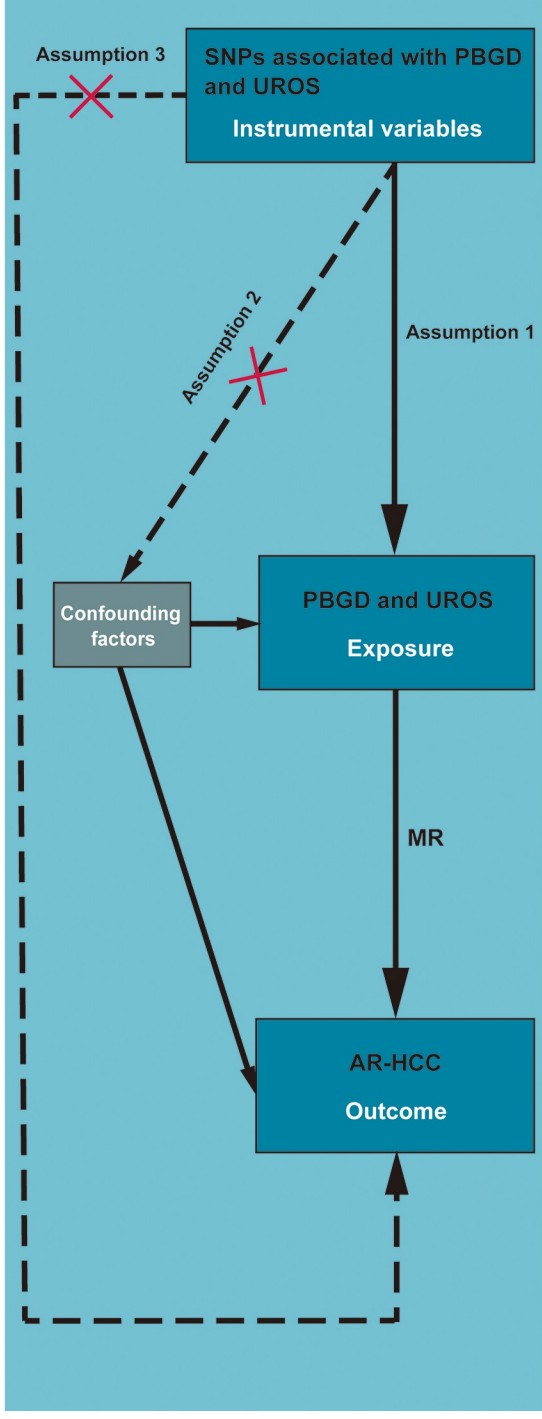

**Fig 1. The study design of this MR study.**

### Data sources

The MR analysis utilized summary-level data from publicly available genome-wide association studies (GWASs). In particular, the GWAS summary statistics for PBGD (GWAS ID: prot-a-1348, n = 10,534,735) and UROS (GWAS ID: prot-a-3176, n = 10,534,735) were extracted from 3,301 samples, while those for AR-HCC (GWAS ID: ebi-a-GCST90092003, n = 7,800,543) were extracted from the results of 775 cases and 1,332 non-cases. This study did not require ethical approval as all GWAS data are publicly available and received prior approval by the relevant ethics committees.

### Selection and validation of the SNPs

Three criteria were used to select appropriate SNPs. Initially, we identified SNPs associated with PBGD and UROS at a threshold of $p < 1 \times 10^{-5}$. Second, we evaluated the independence of the SNPs using pairwise linkage disequilibrium [23]. SNPs with $r^2 > 0.001$ (within a clumping window of 10,000 kb) were excluded if highly correlated with more SNPs or if they had a higher p-value. Third, the strength of the individual SNPs was determined by calculating the F-statistic ($F = Beta^2/Se^2$). SNPs with F-statistics greater than 10 were considered sufficiently strong to mitigate potential bias [24, 25]. PhenoScanner V2 (http://www.phenoscanner.medschl.cam.ac.uk/) [26] is a human genotype-phenotype association database that is used to filter out inappropriate SNPs. After examining the traits of each SNP using the PhenoScanner website, the SNPs that exhibited significant associations with potential confounding factors were excluded. Prior to conducting MR analysis, we performed data-harmonization steps to ensure that the effects of an SNP on both exposure and outcome were aligned with the same allele. A schematic of the MR analysis is presented in Fig 1.

### MR analysis

Five methods were used to assess the causal impact of exposure on the outcome: inverse variance-weighted (IVW), MR-Egger, weighted median, simple mode, and weighted mode. The IVW method provides accurate estimates in the absence of a horizontal pleiotropic balance and weak instrumental bias [27]. In this study, the IVW method was used as the primary method. The MR-Egger method was used to assess the horizontal pleiotropy of selected instrumental variables (IVs) [28]. The weighted median method provides reliable estimates when more than 50% of the information originates from valid IVs [29]. In addition, this simple mode ensures robustness against pleiotropy [30]. The weighted mode is sensitive to challenging bandwidth selections for mode estimation [31]. Cochrane's Q-value test was performed to determine the heterogeneity among the selected IVs. Additionally, a leave-one-out sensitivity analysis was carried out to detect the effect of missing instrumental variables on the remaining instrumental variables. To ensure a more rigorous interpretation of the causal effect, we used a Bonferroni-corrected threshold based on the number of biomarkers ($p < 0.025$, 0.05/2). A nominal causal effect was considered if the p-value was between 0.025 and 0.05. All our statistical analyses were performed using the "TwoSampleMR" packages in R version 4.3.1 (R Foundation for Statistical Computing, Vienna, Austria).

## Results

### SNP selection and validation

All studies included in this MR analysis were published between 2018 and 2021 from the same European ancestry (S1 Table). The GWAS datasets for PBGD (GWAS ID: prot-a-1348) and UROS (GWAS ID: prot-a-3176) were obtained from a previous report [19]. After screening all

**Table 1. MR results of the effects of PBGD and UROS on AR-HCC.**

| Exposure | SNP, N | Method | Effect estimate | 95%CI | P |
|---|---|---|---|---|---|
| PBGD | 14 | Inverse variance weighted | 1.509 | 1.080–2.110 | **0.016** |
| | | MR Egger | 0.688 | 0.249–1.906 | 0.486 |
| | | Weighted median | 1.601 | 0.970–2.641 | 0.066 |
| | | Simple mode | 1.680 | 0.698–4.044 | 0.268 |
| | | Weighted mode | 1.640 | 0.704–3.818 | 0.272 |
| UROS | 10 | Inverse variance weighted | 1.533 | 1.077–2.183 | **0.018** |
| | | MR Egger | 0.918 | 0.344–2.447 | 0.869 |
| | | Weighted median | 1.652 | 1.029–2.652 | **0.038** |
| | | Simple mode | 1.795 | 0.750–4.295 | 0.221 |
| | | Weighted mode | 1.828 | 0.886–3.771 | 0.137 |

MR, Mendelian randomization; PBGD, porphobilinogen deaminase; UROS, uroporphyrinogen-III synthase; AR-HCC, alcohol-related hepatocellular carcinoma; SNP, single-nucleotide polymorphism.

Bold values indicate statistical significance (p < 0.05).

SNPs using PhenoScanner and excluding inappropriate SNPs, all IVs had an F-statistic greater than 10, indicating the absence of weak instrumental bias (S2 Table).

## PBGD and AR-HCC

These results suggest that PBGD has a causal effect on AR-HCC. The effect estimate from the IVW was 1.509 (95% CI:1.080–2.110, p = 0.016; Table 1). A similar significant correlation was not observed across MR-Egger (effect estimate = 0.688, 95% CI = 0.648–1.397, p = 0.486), weighted median (effect estimate = 1.601, 95% CI = 0.970–2.641, p = 0.066), simple mode (effect estimate = 1.680, 95% CI = 0.698–4.044, p = 0.268), and weighted mode (effect estimate = 1.640, 95% CI = 0.704–3.818, p = 0.272) in Table 1. Although part of the analysis did not yield a statistically significant correlation, most of the analysis directions aligned with the IVW analysis. Scatter and forest plots of the association between PBGD and AR-HCC are shown in Figs 2 and 3, respectively. The results of the heterogeneity (MR Egger-Cochrane's Q,

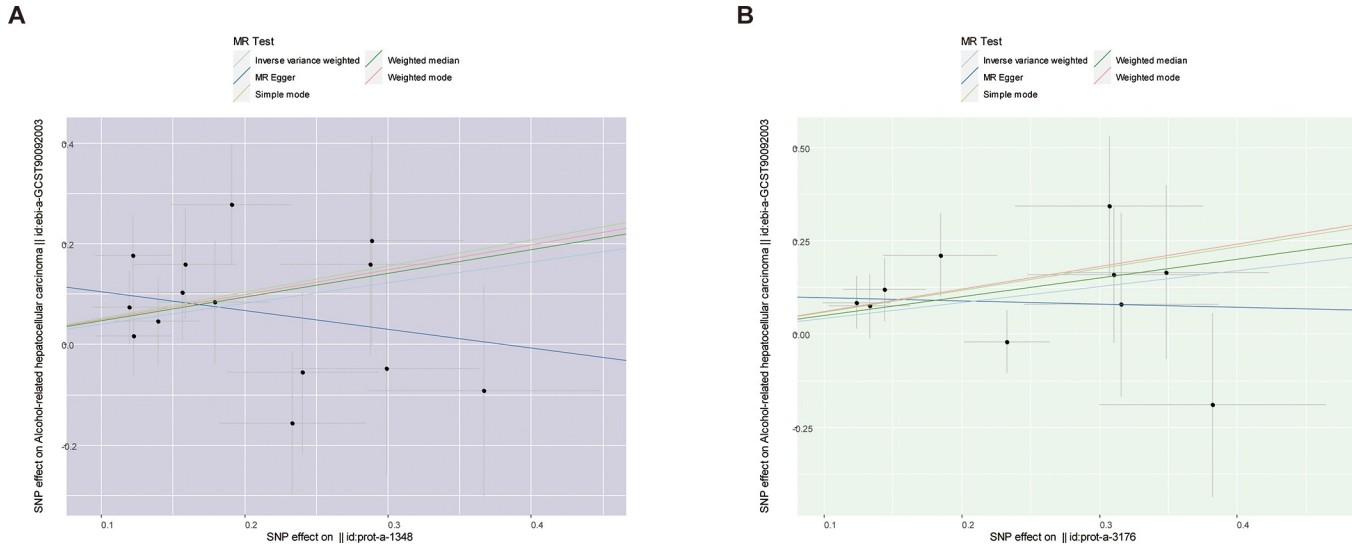

**Fig 2. Scatter plots of the association between porphyria biomarkers and AR-HCC.**

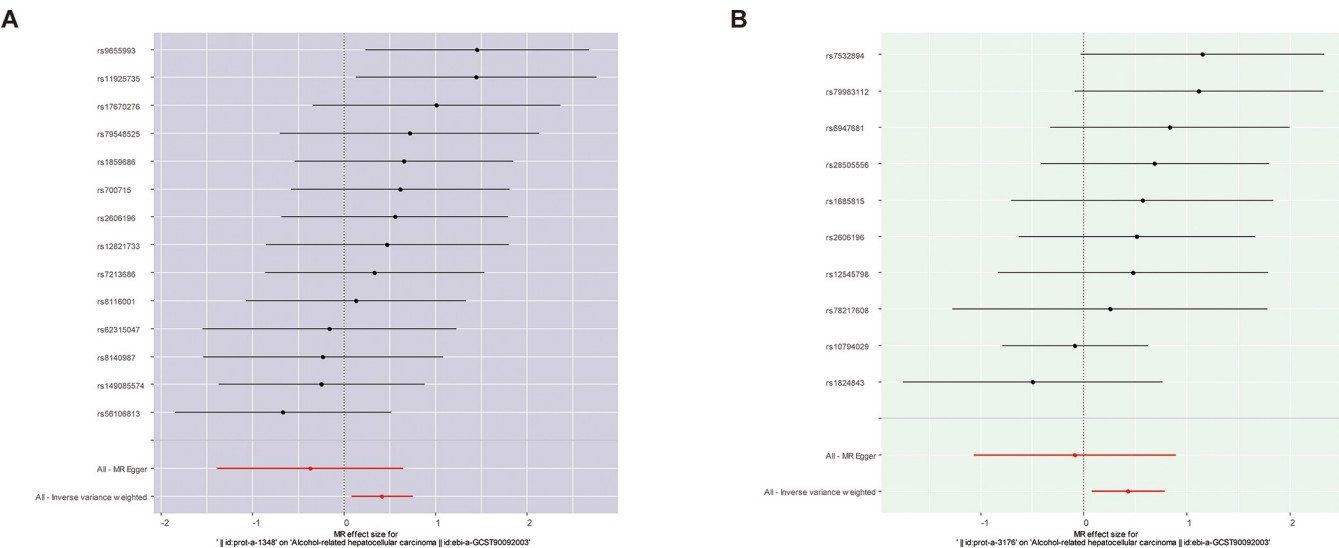

**Fig 3. Forest plots of the association between porphyria biomarkers and AR-HCC.**

p = 0.601; IVM Cochrane's Q, p = 0.469) and pleiotropy tests (p = 0.137) indicate the absence of potential heterogeneity and no evidence of horizontal pleiotropy (Table 2). A funnel plot was used to illustrate the distribution balance of the single SNP effects (S1 Fig). The leave-one-out analysis revealed that no single SNP had a major impact on the final correlation (Fig 4).

## UROS and AR-HCC

The IVW method yielded a significant association between UROS and AR-HCC (effect estimate = 1.533, 95% CI = 1.077–2.183, p = 0.018), and a relative consistent result was obtained from weighted median analysis (effect estimate = 1.652, 95% CI = 1.029–2.652, p = 0.038). The results of the MR-Egger test (effect estimate = 0.918, 95% CI = 0.344–2.447, p = 0.869), simple model (effect estimate = 1.795, 95% CI = 0.750–4.295, p = 0.221), and weighted model (effect estimate = 1.828, 95% CI = 0.886–3.771, p = 0.137) revealed no significant correlation (Table 1). The IVW, weighted median, simple mode, and weighted mode results aligned in the analysis direction. Scatter and forest plots of the association between UROS and AR-HCC are shown in Figs 2 and 3, respectively. No evidence of heterogeneity (MR Egger Cochran's Q, p = 0.601; IVM Cochran's Q, p = 0.469) and pleiotropy (p = 0.137) was observed (Table 2). A funnel plot was used to illustrate the distribution balance of the single SNP effects (S1 Fig). Leave-one-out analysis revealed that the overall estimates were not disproportionately affected by any individual SNP (Fig 4).

**Table 2. Heterogeneity and pleiotropy results of the MR analysis.**

| Exposure | MR Egger Cochrane's Q | | Inverse variance weighted Cochrane's Q | | Pleiotropy | |
|---|---|---|---|---|---|---|
| | Q | P | Q | P | Intercept | P |
| PBGD | 10.172 | 0.601 | 12.731 | 0.469 | 0.142 | 0.136 |
| UROS | 6.399 | 0.603 | 7.607 | 0.574 | 0.107 | 0.304 |

MR, Mendelian randomization; PBGD, porphobilinogen deaminase; UROS, uroporphyrinogen III synthase.

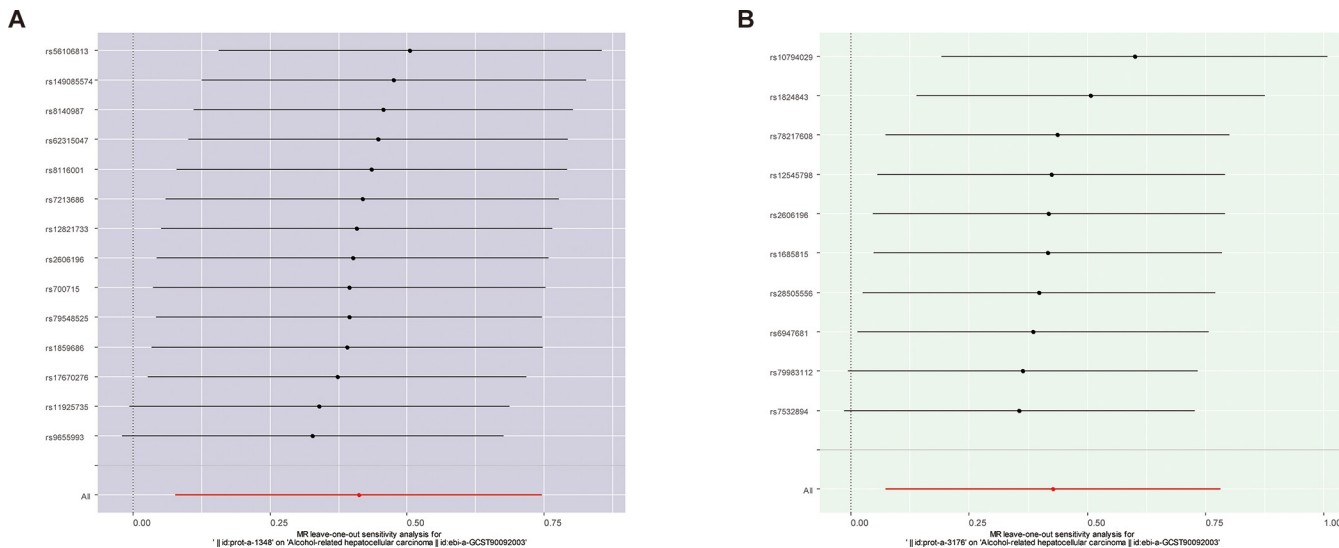

**Fig 4. The leave-one-out analysis of each SNP in porphyria biomarkers and AR-HCC.**

## Discussion

We conducted MR analysis of the relationship between porphyria biomarkers and AR-HCC. Our findings indicate a significant causal effect of PBGD and UROS on AR-HCC.

The MR analysis results revealed a causal correlation between PBGD, a vital biomarker of AIP, and HCC. Several previous studies revealed an association between AIP and HCC [17, 32]; this conclusion is supported by those results. Most previous studies on the association between porphyria and HCC focused on AIP, neglecting other porphyria subtypes. In contrast, our study highlighted a causal association between UROS and AR-HCC. As UROS insufficiency is crucial for CEP, our findings strongly suggest a significant association between CEP and the risk of HCC. The validity and reliability of our MR analysis were supported by the results of the pleiotropy and heterogeneity tests.

This causal relationship may be explained by the mechanisms described below. The accumulation of toxic porphyrin precursors leads to oxidative stress and hepatocyte injury, which are crucial factors in HCC progression [33–35]. Hepatic enzyme deficiency caused by the dysregulation of porphyrin synthesis may lead to liver dysfunction and increase the risk of HCC [36, 37]. Hematopoietic stem cell transplantation has proven to be effective for the early and severe form of CEP [38]. However, treatment-induced hepatitis is a risk factor for HCC development.

Our study had several notable advantages. First, it is a pioneering MR study on the causal effect of porphyria biomarkers and AR-HCC and is the first to address the relationship between CEP and HCC. Second, we effectively minimized the influence of environmental confounding factors and reverse causality through MR analysis, ensuring the reliability of our findings and leading to a more robust conclusion regarding causality. Third, we fulfilled all three assumptions of our MR analysis and established a solid methodological foundation to further reduce the possibility of bias. In addition, all GWAS data included in our study were from European populations. Suitable datasets were included to minimize potential population bias.

Our study had some limitations. First, our analysis was based solely on European populations. Further research in other regions is needed to eliminate racial discrepancies. Second, the

effect of sex was not assessed owing to the limitations of the GWAS dataset. Third, acquiring genomic data and cohort study results for rare diseases is impeded by sample size restrictions. Also, present GWAS datasets regarding PBGD and UROS did not record whether cases were symptomatic or had AR-HCC. On the other hand, the AR-HCC dataset did not clarify whether any of these cases also had porphyria. Future investigations are needed to delve deeper into the relationship between porphyria and HCC.

Our MR analysis revealed a causal relationship between porphyria biomarkers and AR-HCC. This discovery provides novel insights into risk assessment of HCC in patients with porphyria. Furthermore, our findings may hold significance for the discovery of new therapeutics for porphyria and early prevention strategies for HCC.

## Supporting information

**S1 Fig.**
(TIF)

**S1 Table. List of GWAS data included in the MR study.**
(DOCX)

**S2 Table. List of SNPs used as IVs in the MR study of biomarkers.**
(DOCX)

## Acknowledgments

The authors have no relevant financial or non-financial interests to disclose.

## Author Contributions

**Conceptualization:** Xiaoyu Yang, Yunhong Xia.

**Data curation:** Xiaoyu Yang, Shuomin Wang.

**Formal analysis:** Xiaoyu Yang, Chen Sun.

**Investigation:** Xiaoyu Yang, Shuomin Wang, Chen Sun, Yunhong Xia.

**Methodology:** Xiaoyu Yang.

**Software:** Xiaoyu Yang.

**Supervision:** Xiaoyu Yang.

**Validation:** Xiaoyu Yang.

**Visualization:** Xiaoyu Yang.

**Writing – original draft:** Xiaoyu Yang.

**Writing – review & editing:** Xiaoyu Yang, Shuomin Wang, Yunhong Xia.

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
