## [Decision Letter · Decision Letter 0]

15 Jan 2024

PONE-D-23-37115Casual effect of porphyria biomarkers on alcohol-related hepatocellular carcinoma through Mendelian RandomizationPLOS ONE

Dear Dr. Xia,

Thank you for submitting your manuscript to PLOS ONE. After careful consideration, we feel that it has merit but does not fully meet PLOS ONE’s publication criteria as it currently stands. Therefore, we invite you to submit a revised version of the manuscript that addresses the points raised during the review process.

We look forward to receiving your revised manuscript.

Kind regards,

Ashwani Singal

Academic Editor

PLOS ONE

Journal Requirements:

"This study was funded by the National Natural Science Foundation of China (No. 81472331), Natural Science Foundation of Anhui Province (No. 2108085MH289), and the Project of Scientific Research Foundation of Anhui Medical University (No. 2019xkj146)."

Reviewers' comments:

Reviewer's Responses to Questions

**Comments to the Author**

1. Is the manuscript technically sound, and do the data support the conclusions?

Reviewer #1: Partly

Reviewer #2: Yes

2. Has the statistical analysis been performed appropriately and rigorously? 

Reviewer #1: I Don't Know

Reviewer #2: Yes

3. Have the authors made all data underlying the findings in their manuscript fully available?

Reviewer #1: Yes

Reviewer #2: Yes

4. Is the manuscript presented in an intelligible fashion and written in standard English?

Reviewer #1: Yes

Reviewer #2: Yes

5. Review Comments to the Author

Reviewer #1: 1. Novelty: this study fulfils criteria for novelty.

2. Biological plausibility: The acute hepatic porphyrias are associated with a risk of primary liver cancer especially among older patients of 50 years with active porphyria. Previous studies (ref 14 & 15) have shown that risk of primary liver cancer is increased in symptomatic patients with Acute Intermittent Porphyria (AIP) due to accumulation of porphyrin precursors.

3. Methodology: The authors have used a novel technique called Mendelian Randomization-a research method that provides evidence about putative causal relations between modifiable risk factors and disease, using genetic variants as natural experiment. This methodology is less likely to be affected by biases such as confounding or reverse causation.

4. Conclusion: Conclusion is misleading as nowhere in the manuscript there is description of cases with alcohol-related hepatocellular carcinoma (AR-HCC).

comments:

5. Spelling correction for manuscript title on line 4. It should be “Causal” instead of Casual.

6. Spelling correction on line 26 “porphyria”

7. The authors should remove statement in lines 83-85 of the introduction section, which implies that they have already concluded their hypothesis even before presenting the data.

8. Increased risk of primary liver cancer in AIP and other hepatic porphyrias is well-documented, however it is unclear if congenital erythropoietic porphyria (CEP), an even rare form of cutaneous porphyria, is associated with primary liver cancer. What prompted the authors to look for UROS SNPs in CEP cases?

9. Mechanism of liver injury and progression to AR-HCC in porphyrias was briefly mentioned, it may be worthwhile to expand on it.

10. The authors write that “the causal effect of PBGD and UROS on AR-HCC were confirmed using MR”, as the title also suggests. However, nowhere the causes of HCC are described. If anything, this study establishes a causal associated with HCC but not specifically AR-HCC.

11. Previous studies reported increased urinary PBD in symptomatic patients with AIP and incident cases of AR-HCC. It is unclear from this study whether GWAS cases were symptomatic or had AR-HCC. If that is case, it needs to be stated clearly.

12. Also unclear is why there is increased risk of AR-HCC as opposed to other causes of HCC. This needs to be explained in introduction.

Reviewer #2: In this study, Dr. Xia et al. investigated causal relationships between single-nucleotide polymorphisms (SNPs) associated with porphobilinogen deaminase (PBGD) and uroporphyrinogen-III synthase (UROS), and alcohol related HCC (ARHCC) using the public genome-wide association studies (GWAS). The results found that both PBGD (effect estimate = 1.51; 95% CI, from 1.08 to 2.11, p = 0.016) and UROS (effect estimate = 1.53; 95% CI, from 1.08 to 2.18, p = 0.018) have a significant causal effect on AR-HCC. The authors concluded that both AIP and CEP have a causal association with AR-HCC1.

The study investigates a fairly novel concept and technqiue, but there are several limitations which need to be addressed. These are:

Abstract should clarify how many HCC and non-HCC cases extracted the data on SNP for the two genes of heme metabolism.

The authors should clarify what formed their basis for the porphyria genes f0r their association with AR-HCC. This should be clarified in the introduction of the manuscript and purpose of the abstract section.

It should be clarified whether any of these cases also had porphyria (biochemical and/or symptomatic).

Further, the manuscript should be read by an expert in English language for syntax and grammatical errors.

6. PLOS authors have the option to publish the peer review history of their article (what does this mean?). If published, this will include your full peer review and any attached files.

Reviewer #1: No

Reviewer #2: **Yes: **Ashwani K. Singal

---

## [Author Response · Author response to Decision Letter 0]

9 Feb 2024

Dear Dr Singal, 

Plos One

Thank you very much for giving us an opportunity to resubmit the enclosed manuscript entitled “Casual effect of porphyria biomarkers on alcohol-related hepatocellular carcinoma through Mendelian Randomization” (PONE-D-23-37115) for possible publication in Plos One. 

On behalf of my coauthors, I would like to thank you and the reviewers for your valuable suggestions and comments. I have carefully incorporated all comments of reviewers into the newly revised manuscript. All major changes are in red in this revised version. 

Below are my responses to the reviewers’ comments. I believe all concerns have been addressed, and the quality of the revised manuscript has been improved significantly. I sincerely hope the revised manuscript will be acceptable to you and the reviewers. If you have any further questions about the manuscript, please feel free to contact me. Your consideration for this manuscript to be published in Plos One is much appreciated. 

Best regards,

Yunhong Xia

Department of Oncology, the First Affiliated Hospital of Anhui Medical University, Hefei, Anhui, China.

Email: yhxia21@sina.com

Journal Requirements

Response: Thank you for your attention to detail. We have carefully reviewed the requirement and made the necessary changes to the names of supporting information figures and tables in accordance with the MANUSCRIPT BODY FORMATTING GUIDELINES. Additionally, we have addressed several other minor errors that were present in the manuscript.

Response: Thank you for your valuable information. We believe that all the data relevant to our study are already available and relatively accessible. Nevertheless, we will certainly consider the idea of using a repository for our future research. Again, we appreciate your valuable suggestion.

"This study was funded by the National Natural Science Foundation of China (No. 81472331), Natural Science Foundation of Anhui Province (No. 2108085MH289), and the Project of Scientific Research Foundation of Anhui Medical University (No. 2019xkj146)."

Response: Your suggestion is well received. Now we would like to state that the funders had no role in study design, data collection and analysis, decision to publish, or preparation of the manuscript.

Response: Your suggestion is well received. Now we would like to confirm that our submission contains all raw data required to replicate the results of your study.

Review Comments to the Author

Reviewer #1: 

comments:

Spelling correction for manuscript title on line 4. It should be “Causal” instead of Casual.

Response: Thank you very much. We are very sorry for our negligence of this typo. The mistake has been corrected. Please see line 4.

Spelling correction on line 26 “porphyria”

Response: Thank you very much for your thorough review. We apologize for the incorrect writing. The error has been corrected. Please see line 27.

The authors should remove statement in lines 83-85 of the introduction section, which implies that they have already concluded their hypothesis even before presenting the data.

Response: Thank you very much for your helpful suggestion. We have removed this statement in the original line 83-85.

Increased risk of primary liver cancer in AIP and other hepatic porphyrias is well-documented, however it is unclear if congenital erythropoietic porphyria (CEP), an even rare form of cutaneous porphyria, is associated with primary liver cancer. What prompted the authors to look for UROS SNPs in CEP cases?

Response: Thank you for your question. CEP is a result of an error in heme synthesis caused by UROS. Currently, there is limited research on the relationship between CEP and AR-HCC. Therefore, we hypothesized that CEP biomarkers SNPs could serve as a potential entry point for further investigation. To conduct our study, we utilized preexisting GWAS dataset that provided data on UROS SNPs. 

Mechanism of liver injury and progression to AR-HCC in porphyrias was briefly mentioned, it may be worthwhile to expand on it.

Response: Thank you very much for your constructive and helpful suggestion. The mechanism of liver injury and progression to AR-HCC in porphyrias remains poorly understood due to limited studies. However, this also indicates that our research is somewhat innovative. We hope that our findings will help overcome barriers on this topic. Despite the limited current research, we have added some contents regarding AR-HCC and porphyrias. Please see line 59-61 and line 77-78.

The authors write that “the causal effect of PBGD and UROS on AR-HCC were confirmed using MR”, as the title also suggests. However, nowhere the causes of HCC are described. If anything, this study establishes a causal associated with HCC but not specifically AR-HCC.

Response: Thank you very much. The trait of the outcome GWAS dataset (ebi-a-GCST90092003) in our study is documented as AR-HCC (PMID: 34902334). The methodological basis of our conclusion was provided. Additionally, MR research is often restricted to the population region of the exposure and outcome data. Currently, GWAS research on porphyria biomarkers primarily focuses on the European population, while research on HCC still requires further expansion in the future. Therefore, we are also eagerly anticipating the availability of more high quality GWAS data concerning HCC, which would greatly benefit our future study. Please see line 109-111.

Previous studies reported increased urinary PBD in symptomatic patients with AIP and incident cases of AR-HCC. It is unclear from this study whether GWAS cases were symptomatic or had AR-HCC. If that is case, it needs to be stated clearly.

Response: Thank you for your valuable suggestion. The PBGD and UROS datasets related to our study were extracted from a genomic atlas of the human plasma proteome study, without records of AR-HCC cases numbers (PMID: 29875488). We have mentioned it in the revised manuscript. Please see line 229-233. Although the absence of this aspect does not impact the process of reaching the conclusions of this study, it is our hope that there will be an increase in the number of high-quality studies in the future to enhance research in this particular field.

Also unclear is why there is increased risk of AR-HCC as opposed to other causes of HCC. This needs to be explained in introduction.

Response: Thank you very much for your helpful suggestion. Up to today, the GWAS dataset of HCC in European population is focused on AR-HCC only. Due to limited research, future research on other causes of HCC is needed to expand on it. Please see line 79-83.

Special thanks to you for your good comments.

Reviewer #2: 

The study investigates a fairly novel concept and technique, but there are several limitations which need to be addressed. These are:

Abstract should clarify how many HCC and non-HCC cases extracted the data on SNP for the two genes of heme metabolism.

Response: Thank you very much for your helpful suggestion. The PBGD and UROS datasets in our study were extracted from a genomic atlas of the human plasma proteome study, without records of AR-HCC cases numbers (PMID: 29875488). We have mentioned it in the revised manuscript. Please see line 239-243. Although this lack does not affect the process of reaching the conclusions of this study, we hope that further high-quality research can be conducted in the future to strengthen research in this specific area.

The authors should clarify what formed their basis for the porphyria genes for their association with AR-HCC. This should be clarified in the introduction of the manuscript and purpose of the abstract section.

Response: Thank you very much for your helpful suggestion. Former studies showed a possible association between AIP and liver cancer was identified through investigation on porphyria biomarkers. PBGD and UROS are important biomarkers of AIP and CEP, respectively. GWAS studies on porphyria biomarkers are still in the emerging stage, however, high-quality data of PBGD and UROS is already available. We have rewritten these sections following your valuable suggestion. Please see line 28-29 and line 73-83.

It should be clarified whether any of these cases also had porphyria (biochemical and/or symptomatic).

Response: Thank you very much for your constructive suggestion. The original study did not record whether these cases also had porphyria (PMID: 34902334). We have mentioned it in the revised manuscript. Please see line 239-243. Again, we hope that high-quality research can be conducted in the future to improve research in this specific area.

Further, the manuscript should be read by an expert in English language for syntax and grammatical errors.

Response: Thank you very much for your helpful suggestion. We invited native English speakers to polish the language of this paper. The wrong words have been removed in revised version.

Special thanks to you for your kind comments.

---

## [Editor Report · Decision Letter 1]

13 Feb 2024

Casual effect of porphyria biomarkers on alcohol-related hepatocellular carcinoma through Mendelian Randomization

PONE-D-23-37115R1

Dear Dr. Xia,

We’re pleased to inform you that your manuscript has been judged scientifically suitable for publication and will be formally accepted for publication once it meets all outstanding technical requirements.

Kind regards,

Ashwani Singal

Academic Editor

PLOS ONE

Additional Editor Comments (optional):

Authors have appropriately revised the paper with point by point responses to the reviewers comments.

No further comments from this reviewer.
---

## [Editor Report · Acceptance letter]

11 Mar 2024

PONE-D-23-37115R1 

PLOS ONE

Dear Dr. Xia, 

I'm pleased to inform you that your manuscript has been deemed suitable for publication in PLOS ONE. Congratulations! Your manuscript is now being handed over to our production team.

Kind regards, 

on behalf of

Dr. Ashwani Singal 

Academic Editor

PLOS ONE